# FASTER SINKHORN'S ALGORITHM WITH SMALL TREEWIDTH

## ABSTRACT

While approximating optimal transport (OT) distances such as the earth mover's distance is a fundamental problem in statistics and machine learning, it is computationally expensive. Given the cost matrix $C = AA^\top$ where $A \in \mathbb{R}^{n \times d}$, the state-of-the-art results [Dvurechensky, Gasnikov, and Kroshnin ICML 2018] cost $\widetilde{O}(\epsilon^{-2} n^2)$ time to approximate OT distance, where $n$ is the size of given two discrete distributions and $\epsilon$ is the error. In this paper, we proposed a faster Sinkhorn's Algorithm to approximate the OT distance when matrix $A$ has treewidth $\tau$, which is usually very small. Our algorithm achieves a running time of $\widetilde{O}(\epsilon^{-2} n \tau)$, improving upon the previous $\widetilde{O}(\epsilon^{-2} n^2)$ time complexity. To the best of our knowledge, our paper is the first work to improve the OT distance approximating problem running time to $\widetilde{O}(\epsilon^{-2} n \tau)$.

## 1 INTRODUCTION

Optimal transport is a mathematical theory that deals with the problem of finding the most efficient way to transport goods or materials from one place to another. The goal is to minimize the cost of transportation, which is usually measured in terms of the distance traveled or the amount of resources used. Many problems in computational sciences require to use optimal transport to compare probability measures or histograms, including Wasserstein or earth mover's distance (Werman et al., 1985; Rubner et al., 2000; Villani, 2009). Optimal transport has a wide range of applications, such as bag-of-words for natural language processing (Kusner et al., 2015), multi-label classification (Frogner et al., 2015), unsupervised learning (Arjovsky et al., 2017; Bigot et al., 2017), semi-supervised learning (Solomon et al., 2014), statistics (Ebert et al., 2017; Panaretos & Zemel, 2016), and other application (Kolouri et al., 2017). In particular, due to its applications in image processing, it has recently become crucial to have efficient ways of computing, or approximating, the optimal transport or the Wasserstein distances between two measures.

Optimal Transport (OT) problems have been the focus of extensive research. One significant advancement in this field came with the application of Sinkhorn's algorithm to entropy-regularized OT optimization, as highlighted in (Cuturi, 2013). This application proved beneficial in tackling the OT challenge. As it was recently shown in (Altschuler et al., 2017), this approach allows to find an $\epsilon$-approximation for an OT distance in $\widetilde{O}(\epsilon^{-3} n^2)$ time. In terms of the dependence on $n$, this result improves on the complexity $\widetilde{O}(n^3)$ achieved by the network simplex method or interior point methods (Pele & Werman, 2009), applied directly to the OT optimization problem, which is a linear program (Kantorovich, 1942). The cubic dependence on $\epsilon$ prevents approximating OT distances with good accuracy. Then, in (Dvurechensky et al., 2018), they proposed an algorithm with the complexity bound $\widetilde{O}(\epsilon^{-2} n^2)$ based on the Sinkhorn's algorithm.

The treewidth of a matrix is a measure of the complexity of its structure and plays a crucial role in the design and analysis of algorithms for manipulating and processing matrices. In particular, the treewidth of a matrix can be used to determine the efficiency of algorithms that rely on tree decompositions, such as dynamic programming and divide-and-conquer techniques. In the small treewidth setting, algorithms for matrix manipulation and processing can often achieve near-linear running time, making them highly efficient and scalable. This has important implications for a wide range of applications, including interior point methods (Gu & Song, 2022), computing John ellipsoid (Song et al., 2022), streaming algorithm (Liu et al., 2022). Treewidth is also important in graph

structure theory, particularly in the study of graph minors by Robertson and Seymour (Robertson & Seymour, 2010). Many results (Bodlaender, 1994) have shown that NP-hard problems can be solved in polynomial time on classes of graphs with bounded treewidth.

The best previous work to solve this problem requires $n^2$. It is natural to ask a question

*Is that possible to solve in $o(n^2)$ under some mild assumption, e.g. treewidth?*

In this paper, we provide a positive answer for this question. The comparison between our results and previous works is shown in Table 1.

Table 1: Given the cost matrix $C = AA^\top \in \mathbb{R}^{n \times n}$, let $\tau$ denote the treewdith of matrix $A$. Let $\epsilon$ denote the accuracy parameter. Since $\tau \leq n$, our algorithm (Theorem B.3, Algorithm 1) is always better than (Dvurechensky et al., 2018).

| References | Method | Time Complexity |
|---|---|---|
| (Pele & Werman, 2009) | Network Simplex Method | $n^3$ |
| (Altschuler et al., 2017) | Sinkhorn's algorithm | $\epsilon^{-3}n^2$ |
| (Dvurechensky et al., 2018) | Sinkhorn's algorithm | $\epsilon^{-2}n^2$ |
| Theorem B.3 | Sinkhorn's algorithm | $\epsilon^{-2}n\tau$ |

## 1.1 OUR RESULT

We formally state our main theorem

**Theorem 1.1.** *Given the cost matrix $C = AA^\top$ where $A$ has treewidth $\tau$, we can find the transport plan for the $\epsilon$-approximation of the optimal transport distance in $O(\epsilon^{-2}n\tau\|C\|_\infty^2 \ln n)$ time.*

Comparing with (Dvurechensky et al., 2018), which solves the problem in $O(\epsilon^{-2}n^2\|C\|_\infty^2 \ln n)$, we proposed an algorithm that constructs a matrix using its implicit form. By leveraging the property of low treewidth, our running time has no dependence on $n^2$.

## 1.2 RELATED WORK

**OT Problems** OT distances, which is also called Earth Mover's Distances (Rubner et al., 2000), are progressively being adopted as an effective tool in a wide range of situations, from computer graphics (Bonneel et al., 2016) to supervised learning (Frogner et al., 2015), unsupervised density fitting (Bassetti et al., 2006) and generative model learning ((Montavon et al., 2016; Arjovsky et al., 2017; Salimans et al., 2018; Genevay et al., 2018; Sanjabi et al., 2018)). There is a long line of work on reducing the time complexity for solving OT. In (Arjovsky et al., 2017), they proved that, for regularized OT, the near-linear time complexity can be achieved by both Sinkhorn and Greenkhorn algorithm. They demonstrated that both algorithms have a complexity of $\widetilde{O}(\epsilon^{-3}n^2)$, where $n$ represents the number of atoms (or the dimension) of the probability measure being considered and $\epsilon$ is the desired level of tolerance. In (Dvurechensky et al., 2018), the complexity of the Sinkhorn algorithm was improved to $\widetilde{O}(\epsilon^{-2}n^2)$. Additionally, an adaptive primal-dual accelerated gradient descent (APDAGD) algorithm was introduced, which was shown to have a complexity of $\widetilde{O}(\min\{\epsilon^{-1}n^{9/4}, \epsilon^{-2}n^2\})$. With a carefully designed Newton-type algorithm, (Allen-Zhu et al., 2017; Cohen et al., 2017) solve the OT problem by making use of a connection to matrix-scaling problems. (Blanchet et al., 2018; Quanrud, 2018) gave a complexity bound of $\widetilde{O}(\epsilon^{-1}n^2)$ for Newton-type algorithms.

**Treewidth Problems** Treewidth is a concept from structural graph theory that has been studied in relation to fixed-parameter tractable algorithms in various fields, including combinatorics, integer-linear programming, and numerical analysis. In practical settings, treewidth tends to be small. A study by (Zhang & Lavaei, 2021) on the MATPOWER data set for power system analysis revealed that the largest problem size was ($n = 12659, m = 20467$), with a maximum treewidth of $\tau = 35$. As a result, it is reasonable to conclude that treewidth-efficient algorithms surpass general-purpose matrix algorithms in practical applications. (Fomin et al., 2018) shows several problems

can be reduced to matrix factorizations efficiently, including computing determinant, computing rank, and finding maximum matching, and this leads to $O(\tau^{O(1)} \cdot n)$ time algorithms where $\tau$ is the width of the given tree decomposition of the graph. (Bodlaender, 1994) shows a number of NP-hard problems such as INDEPENDENT SET, HAMILTONIAN CIRCUIT, STEINER TREE, AND TRAVELLING SALESMAN can be solved with run-times that depend only linearly on the problem size and exponentially on treewidth as the result of dynamic programming. By leveraging the small treewidth setting, (Gu & Song, 2022) proposed an algorithm that solves the linear program problem with run-time nearly matching the fastest run-time for solving the sub-problem $Ax = b$. (Liu et al., 2022) proposed a space-efficient interior point method (IPM) in the streaming model. For the linear programs with treewidth $\tau$, they solve them in $\widetilde{O}(n\tau)$ space, where $n$ is the number of dimensions for the feature space. (Song et al., 2022) shows that, when the constraints matrix has treewidth $\tau$, the John Ellipsoid problem can be solved in $O(n\tau^2)$ time. The small treewidth setting is also applied to solve the semidefinite program. In (Gu & Song, 2022), they give the first SDP solver that runs in time in linear in a number of variables under this setting. In (Gu et al., 2023), they study the linear support vector machine problem and kernel support vector machine problem. They provide the first nearly linear time algorithm for solving the SVM via interior point method.

### 1.3 TECHNIQUE OVERVIEW

**Analysis**  We first provide preliminaries, which include necessary notations, problem formulation and treewidth basics, as well as some definitions used in Sinkhorn algorithm. After that, we introduce the convex function of $(\widehat{u}, \widehat{v})$ as the following: $\langle \mathbf{1}_n, B(\widehat{u}, \widehat{v})\mathbf{1}_n \rangle - \langle \widehat{u}, B(u_k, v_k)\mathbf{1}_n \rangle - \langle \widehat{v}, B(u_k, v_k)^\top \mathbf{1}_n \rangle$. The gradient for the above function vanishes when $(u^*, v^*) = (u_k, v_k)$, so the point $(u_k, v_k)$ is the minimizer of this function. Therefore, we can show that $\widetilde{\psi}(u_k, v_k) \leq \langle u_k - u_*, B_k \mathbf{1}_n - r \rangle + \langle v_k - v_*, B_k^\top \mathbf{1}_n - c \rangle$ Then, for each iteration of the algorithm, we upper bound the r.h.s. and get $\widetilde{\psi}(u_k, v_k) \leq R \cdot (\|B_k \mathbf{1}_n - r\|_1 + \|B_k^\top \mathbf{1}_n - c\|_1)$. where the inequality follows from the bounds for the iterates $u_k, v_k$ and an optimal solution $(u^*, v^*)$. Next, by using this upper bound for $\widetilde{\psi}$ and Pinsker inequality (Lemma B.2) we have:

$$\widetilde{\psi}(u_k, v_k) - \widetilde{\psi}(u_{k+1}, v_{k+1}) \geq \max\{\frac{\widetilde{\psi}(u_k, v_k)^2}{2R^2}, \frac{\epsilon_0^2}{2}\},$$

By using induction, we prove the potential function $\widetilde{\psi}$ is also upper bounded by $\frac{2R^2}{k+\ell-1}$, where $\ell = \frac{2R^2}{\widetilde{\psi}(u_1, v_1)}$. Finally, by using the switching strategy, we provide the upper bound of the total number of iterations $k$ for the Sinkhorn's algorithm as the following $k \leq 2 + \frac{4R}{\epsilon_0}$.

**Running time**  Given the cost matrix $C = MM^\top$ where $M \in \mathbb{R}^{n \times d}$ has treewidth $\tau$, we leverage the fact that it admits a succinct Cholesky factorization and $\mathrm{nnz}(C) = O(n\tau)$, where nnz refers to the number of non-zeros of a matrix.

For each iteration in Sinkhorn's algorithm (Algorithm 3), we have to compute $B(u, v) = \mathrm{diag}(e^u) K \mathrm{diag}(e^v)$ where $K_{i,j} := \exp(-C_{i,j}/\gamma)$. In fact, writing down $K$ explicitly requires $O(n^2)$. To bypass this issue, we first write $K$ in implicit form $K_{i,j} := A_{i,j} - D_{i,j}$. where $A_{i,j} = e^{-C_{i,j}/\gamma} - 1$ and $D_{i,j} = 1$, so that matrix $A$ is as sparse as matrix $C$. Also, we represent matrix $D$ by $ww^\top$, where $w = \mathbf{1}_n$. Leveraging the fact that $\mathrm{nnz}(A) = O(n\tau)$ and matrix $D$ is a rank-1 matrix, we improve the per iteration running time for Sinkhorn algorithm from $O(n^2)$ to $O(n\tau)$.

For the rounding algorithm (Algorithm 2) of the transport plan $B$, we also write down the transport plan in an implicit fashion and do the computation in $O(n\tau)$ time. Note that we *never* write down $B, B_0, B_1$ and output $G$ explicitly. When computing $B\mathbf{1}_n$, we leverage the implicit form of $B$ and do the computation as following:

$$\mathrm{diag}(e^{u_k}) A \mathbf{1}_n \mathrm{diag}(e^{v_k}) + \mathrm{diag}(e^{u_k})(ww^\top)\mathbf{1}_n \mathrm{diag}(e^{v_k}).$$

As $\mathrm{nnz}(A) = O(n\tau)$, computing $A\mathbf{1}_n$ takes $O(n\tau)$ time. Similarly, when computing $XB$, where $X$ is a diagonal matrix, we leverage the implicit form of $B$ and do the computation as following:

$$\mathrm{diag}(e^{u_k}) A X \mathrm{diag}(e^{v_k}) + \mathrm{diag}(e^{u_k})(ww^\top) X \mathrm{diag}(e^{v_k}).$$

As $\mathrm{nnz}(A) = O(n\tau)$ computing $AX$ takes $O(n\tau)$ time and the $AX$ is also $O(n\tau)$ sparse.

Finally, we note that the transport plan for the OT distance problem can be approximated in $\widetilde{O}(\epsilon^{-2}n\tau)$ time.

**Roadmap.** We first introduce all required preliminary in Section 2. Then, we provide the analysis for the Sinkhorn's algorithm in Section 3. In Section 4, we provide a faster Sinkhorn's algorithm with small treewidth setting and apply our faster Sinkhorn's Algorithm to solve the OT distance. In Section 5, we conclude our contribution for this paper.

## 2 PRELIMINARY

### 2.1 NOTATIONS

For a positive integer $n$, we denote $[n] = \{1, 2, \cdots, n\}$ We use $\mathbf{1}_n$ to denote the length-$n$ vector where all the entries that are ones. For a vector $a$, we denote $e^a, \ln a$ as their entry-wise exponents and natural logarithms respectively. We define $a_{k,i}$ as the $i$-th coordinate of $k$-th iteration of $a$. For a matrix $A \in \mathbb{R}^{n \times n}$, we define $\|A\|_\infty := \max_{i,j \in [n]} |A_{i,j}|$. We define $A_{i,j}$ as the entry at $i$-th row and $j$-th column of matrix $A$. We use $e^A, \ln A$ to denote their entry-wise exponents and natural logarithms respectively. We denote by $\mathrm{vec}(A)$ the vector in $\mathbb{R}^{n^2}$, which is obtained from $A$ by writing its columns one below another. For two matrices $A, B$, we denote their inner product by $\langle A, B \rangle$. We define the $n$-dimensional simplex as $\triangle_n := \{x \in \mathbb{R}^n_+ : \sum_{i=1}^n x_i = 1\}$. For a vector $x \in \mathbb{R}^n$, we define its $\ell_p$ norm to be $\|x\|_p := (\sum_{i=1}^n |x_i|^p)^{1/p}$. For two vectors $x, y$, we define the inner product $\langle x, y \rangle = \sum_{i=1}^n x_i y_i$. We use $\mathrm{nnz}(\cdot)$ to denote the number of non-zeros of a matrix.

### 2.2 PROBLEM FORMULATION

The definition of entropy is given as the following:

**Definition 2.1** (Entropy)**.** *We define the entropy $H(p)$ of vector $p$ by $H(p) := \sum_{i=1}^n p_i \log(\frac{1}{p_i})$. Similarly, for a matrix $P \in \mathbb{R}^{n \times n}_+$, we define the entropy $H(P)$ entrywise as*

$$\sum_{i=1}^n \sum_{j=1}^n P_{i,j} \log \frac{1}{P_{i,j}}.$$

We first introduce the definition of OT problem.

**Definition 2.2.** *Given a matrix $C$ with small treewidth (e.g. $C = AA^\top$ where $A \in \mathbb{R}^{n \times d}$), the optimal transport problem is defined as:*

$$\min_X \langle C, X \rangle \quad \text{s.t.} \quad X \in \mathbb{R}^{n \times n}_+, \quad X\mathbf{1}_n = r, \quad X^\top \mathbf{1}_n = c,$$

*where $\mathbf{1}_n \in \mathbb{R}^n$ denotes a vector where every entry is 1.*

Next, we give the definition of the regularized OT problem.

**Definition 2.3.** *Given a strongly convex regularizer $\mathcal{R}(X)$, e.g. negative entropy or squared Euclidean norm, the regularized optimal transport problem is defined as:*

$$\min_X \langle C, X \rangle + \gamma \mathcal{R}(X), \quad \text{s.t.} \quad X \in \mathbb{R}^{n \times n}_+, \quad X\mathbf{1}_n = r, \quad X^\top \mathbf{1}_n = c, \tag{1}$$

*where $\gamma > 0$ denotes the regularization parameter.*

The goal for this paper is to find the approximation for the transportation plan $\widehat{X}$ defined as follows:

**Definition 2.4** ($\epsilon$-approximation)**.** *The $\epsilon$-approximation for the OT distance is defined as*

$$\langle C, \widehat{X} \rangle \leq \min_X \langle C, X \rangle + \epsilon, \quad \text{s.t.} \quad X \in \mathbb{R}^{n \times n}_+, \quad X\mathbf{1}_n = r, \quad X^\top \mathbf{1}_n = c, \tag{2}$$

*where $\widehat{X}$ denotes the approximation for the transportation plan.*

For simplicity we introduce the definition of $\mathcal{U}_{r,c} \subset \mathbb{R}_+^{n \times n}$

**Definition 2.5.** *Given the OT problem* $\arg\min_{X \in \mathcal{U}_{r,c}} \langle X, C \rangle$, *we define* $\mathcal{U}_{r,c} := \{X \in \mathbb{R}_+^{n \times n} : X \mathbf{1}_n = r, X^\top \mathbf{1}_n = c\}$, *where* $\mathbf{1}_n$ *is the all-ones vector in* $\mathbb{R}^n$, $C \in \mathbb{R}_+^{n \times n}$ *is a given cost matrix, and* $r \in \mathbb{R}^n, c \in \mathbb{R}^n$ *are given vectors with positive entries that sum to one.*

Next, we provide a lemma about the transport plan $X$.

**Lemma 2.6** ((Cuturi, 2013)). *For any cost matrix* $C \in \mathbb{R}^{n \times n}$, $\mathcal{U}_{r,c} \subset \mathbb{R}_+^{n \times n}$ *and* $r, c \in \triangle_n$, *the minimization program* $X_\gamma := \arg\min_{X \in \mathcal{U}_{r,c}} \langle X, C \rangle + \gamma \cdot \mathcal{R}(X)$, *where* $\gamma > 0$ *is the regularization parameter and* $\mathcal{R}(X)$ *is a strongly convex regularizer, has a unique minimum at* $X_\gamma \in \mathcal{U}_{r,c}$ *of the form* $X_\gamma = MAN$, *where* $A := \exp(-\frac{1}{\gamma} C)$ *and* $M, N \in \mathbb{R}_+^{n \times n}$ *are both diagonal matrices. The matrices* $(M, N)$ *are unique up to a constant factor.*

## 2.3 TREEWIDTH PRELIMINARIES

We begin by introducing the definition of treewidth for a given matrix.

**Definition 2.7** (Treewidth $\tau$). *Given a matrix* $A \in \mathbb{R}^{n \times d}$, *we construct its graph* $G = (V, E)$ *as follows: The vertex set are columns* $[d]$; *An edge* $(i, j) \in E$ *if and only if there exists* $k \in [n]$ *such that* $A_{k,i} \neq 0, A_{k,j} \neq 0$. *Then, the treewidth of the matrix* $A$ *is the treewidth of the constructed graph. In particular, every column of* $A$ *is* $\tau$-sparse.

Next, we present the definition for Cholesky factorization.

**Definition 2.8** (Cholesky Factorization). *Given a positive-definite matrix* $P$, *there exists a unique Cholesky factorization* $P = LL^\top \in \mathbb{R}^{d \times d}$, *where* $L \in \mathbb{R}^{d \times d}$ *is a lower-triangular matrix with real and positive diagonal entries.*

We also provide the running time of computing the Cholesky factorization.

**Lemma 2.9** ((George et al., 1994)). *Given a positive definite matrix* $M \in \mathbb{R}^{d \times d}$, *we can decompose it by using Cholesky decomposition* $M = LL^\top$ *in time* $\Theta(\sum_{j=1}^d |\mathcal{L}_j|^2)$, *where* $|\mathcal{L}_j|$ *is the number of nonzero entries in the* $j$-th *column of* $L$.

Then, we introduce some results based on the Cholesky factorization of a given matrix with treewidth $\tau$:

**Lemma 2.10** ((Bodlaender et al., 1995; Davis, 2006)). *For any matrix* $A \in \mathbb{R}^{n \times n}$ *with treewidth* $\tau$, *we can compute the Cholesky factorization* $A = LL^\top \in \mathbb{R}^{n \times n}$ *in* $O(n\tau^2)$ *time, where* $L \in \mathbb{R}^{n \times n}$ *is a lower-triangular matrix with real and positive entries.* $L$ *satisfies the property that every column is* $\tau$-sparse.

Next, we show a standard property of treewidth.

**Claim 2.11** ((Gu & Song, 2022; Song et al., 2022; Liu et al., 2022)). *Given* $L = MM^\top$, *where* $M$ *has treewidth* $\tau$ *and* $M \in \mathbb{R}^{m \times n}$, *we have* $\mathrm{nnz}(L) = O(n\tau)$.

*Proof.* We first show that $\mathrm{nnz}(L) = O(m)$. Let $M \in \mathbb{R}^{m \times n}$ denote the adjacency matrix of graph $G = (V, E)$, where $|E(G)| = m, |V(G)| = n$. The Laplacian matrix of graph $G$ is $L = MM^\top$ and it is also defined as $D - A$, where $D$ is the degree matrix and $A$ is the adjacency matrix of graph $G$. As $\mathrm{nnz}(A) = O(m), \mathrm{nnz}(D) = O(n)$ and $m \geq n$, we have

$$\mathrm{nnz}(L) = O(m) + O(n) = O(m). \tag{3}$$

Next, we show that the number of edge $m$ for graph $G$ is bounded by $O(n\tau)$. The maximal graphs with treewidth $\tau$ are the $\tau$-trees which are constructed by starting with a $(\tau+1)$-clique and iteratively adding vertices of degree $\tau$ such that its neighbors form a $\tau$-clique. By counting the edges in the $(\tau+1)$-clique and the edges incident to the $n - \tau - 1$ vertices iteratively added to the $\tau$-tree, the total number of edges in a $\tau$-tree with $n$ vertices is

$$\binom{\tau+1}{2} + \tau(n - \tau - 1) = O(n\tau). \tag{4}$$

Since any graph G with treewidth $\tau$ is a subgraph of a $\tau$-tree, we have $O(n\tau)$ is an upper bound on $|E(G)| = m$. By combining Eq. (3) and Eq. (4), we have $\mathrm{nnz}(L) = O(n\tau)$.

Hence, we complete the proof. $\qquad\square$

# 3 SINKHORN'S ALGORITHM ANALYSIS

In Section 3.1, we provide some definitions used in Sinkhorn algorithm. In Section 3.2, we define the potential function $\widetilde{\psi}$. In Section 3.3, we provide the upper bound of $\widetilde{\psi}$. In Section 3.4, we provide the induction proof for the upper bound of the potential function.

## 3.1 DEFINITIONS

We first introduce some definitions to simplify the derivations.

**Definition 3.1.** *We define matrix function $B : \mathbb{R}^n \times \mathbb{R}^n \to \mathbb{R}^{n \times n}$ as follows: for any given vectors $u, v \in \mathbb{R}^n$ $B(u, v) := \mathrm{diag}(e^u) K \, \mathrm{diag}(e^v)$, where $\mathrm{diag}(a) \in \mathbb{R}^{n \times n}$ is the diagonal matrix with the vector $a \in \mathbb{R}^n$ on the diagonal and $K \in \mathbb{R}^{n \times n}$ is a matrix which is defined as $K_{i,j} := \exp(-C_{i,j}/\gamma)$.*

**Definition 3.2.** *We define function $\psi : \mathbb{R}^n \times \mathbb{R}^n \to \mathbb{R}$ as follows: for any given vectors $u, v \in \mathbb{R}^n$ $\psi(u, v) := \mathbf{1}_n^\top B(u, v) \mathbf{1}_n - \langle u, r \rangle - \langle v, c \rangle$, where $B$ is defined in Definition 3.1.*

We consider the Sinkhorn algorithm (Algorithm 3), which solves the following minimization problem introduced in Lemma 2 of (Cuturi, 2013):

$$\min_{u,v \in \mathbb{R}^n} \psi(u, v), \tag{5}$$

where $\psi$ is defined in Definition 3.2.

Problem Eq. (5) is the dual formulation to Eq. (1) as we choose $\mathcal{R}(X) = -H(X)$.

Here, we show the high-level idea of proving the complexity of Sinkhorn's algorithm. We first show how to get the bounds for $u_k, v_k$ and an optimal solution $(u_*, v_*)$ for Eq. (5). Next, we show that, for each iteration, $\psi(u_k, v_k)$ is upper bounded by $\|B(u_k, v_k)\mathbf{1}_n - r\|_1 + \|B(u_k, v_k)^\top \mathbf{1}_n - c\|_1$.

Eventually, by using the bound of $\psi(u_k, v_k)$, we show our result of the complexity result for Sinkhorn's algorithm.

**Definition 3.3.** *We define $R$ as $R := -\ln(K_{\min} \min_{i,j \in [n]} \{r_i, c_j\})$, where $K_{\min} := \min_{i,j \in [n]} K_{i,j} = e^{-\|C\|_\infty/\gamma}$.*

## 3.2 POTENTIAL FUNCTION $\widetilde{\psi}$

To simplify derivations, we define $\widetilde{\psi}$ as follows:

**Definition 3.4.** *We define $\widetilde{\psi}$ as $\widetilde{\psi}(u, v) := \psi(u, v) - \psi(u_*, v_*)$.*

**Claim 3.5.** *We have $\widetilde{\psi}(u, v) = \langle \mathbf{1}_n, B(u, v)\mathbf{1}_n \rangle - \langle \mathbf{1}_n, B(u_*, v_*)\mathbf{1}_n \rangle + \langle u_* - u, r \rangle + \langle v_* - v, c \rangle$.*

*Proof.* Since $\widetilde{\psi}(u, v) = \psi(u, v) - \psi(u_*, v_*)$ by definition of $\widetilde{\psi}$, the proof follows from the definition of $\psi$. $\qquad\square$

## 3.3 UPPER BOUNDING FOR POTENTIAL FUNCTION

Here, we provide a lemma which will be used later to bound the iteration complexity.

**Lemma 3.6** (Informal version of Lemma A.2). *Let $k \geq 1$ and $u_k, v_k \in \mathbb{R}^n$ be output of Algorithm 3. We denote $B_k := B(u_k, v_k)$. Then, we have*

$$\widetilde{\psi}(u_k, v_k) \leq R \cdot (\|B_k \mathbf{1}_n - r\|_1 + \|B_k^\top \mathbf{1}_n - c\|_1).$$

## 3.4 INDUCTION PROOF FOR THE UPPER BOUND OF THE POTENTIAL FUNCTION

Here, we provide the induction proof for the upper bound of the potential function.

**Lemma 3.7.** *For all $k \geq 1$, $\frac{\widetilde{\psi}(u_k, v_k)}{2R^2} \leq \frac{1}{k + \ell - 1}$, where $\ell := \frac{2R^2}{\widetilde{\psi}(u_1, v_1)}$ and $\widetilde{\psi}$ is defined in Definition 3.2.*

*Proof.* Our proof can be divided into two parts. At first, we consider the correctness of the inequalities above with $k = 1$. Then, inducing over $k > 1$, the proof will be completed.

**Base Case.** For $k = 1$, it holds that

$$\frac{\widetilde{\psi}(u_1, v_1)}{2R^2} = \frac{1}{\ell} = \frac{1}{k + \ell - 1},$$

where, the first step follows from the definition of $\ell$ and the last step follows from $k - 1 = 0$. Hence, we have $\frac{\widetilde{\psi}(u_k, v_k)}{2R^2} \leq \frac{1}{k + \ell - 1}$ for $k = 1$.

**General case** Suppose

$$\frac{\widetilde{\psi}(u_k, v_k)}{2R^2} \leq \frac{1}{k + \ell - 1}. \tag{6}$$

Then we can show

$$\begin{aligned}
\frac{\widetilde{\psi}(u_{k+1}, v_{k+1})}{2R^2} &\leq \frac{\widetilde{\psi}(u_k, v_k)}{2R^2} - \left(\frac{\widetilde{\psi}(u_k, v_k)}{2R^2}\right)^2 \\
&\leq \frac{1}{k + \ell - 1} - \left(\frac{1}{k + \ell - 1}\right)^2 \\
&\leq \frac{1}{k + \ell}, \tag{7}
\end{aligned}$$

where the first step follows from Lemma A.3, and the second step follows from Eq. (6) and the property of function $f(x) = x - x^2$ (which is $f(y) \leq f(z)$ if $y \leq z \leq 1/2$), the last step follows from $\frac{1}{A} - \frac{1}{A^2} \leq \frac{1}{A+1}$ for any integer $A \geq 2$. By induction, the proof is completed. $\square$

## 4 RUNNING TIME WITH SMALL TREEWIDTH SETTING

In Section 4.1, we introduce the implicit form $K$. In Section 4.2, we provide the faster Sinkhorn' Algorithm with small treewidth. In Section 4.3, we show the correctness of our rounding algorithm. In Section 4.4, we show the running time needed for our rounding algorithm. In Section 4.5, we provide the running time for approximating the OT distance by using the faster Sinkhorn's Algorithm.

### 4.1 IMPLICIT FORM OF $K$

---
**Algorithm 1** Approximate OT by Sinkhorn
---
1: **procedure** APPROXOT($\epsilon$)                                                    ▷ Theorem B.3
2:                                                                                          ▷ Accuracy $\epsilon$
3:     $\gamma \leftarrow \frac{\epsilon}{4 \ln n}$
4:     $\epsilon_0 \leftarrow \frac{\epsilon}{8\|C\|_\infty}$
5:                          ▷ Find $\widetilde{r}, \widetilde{c} \in \Delta^n$ s.t. $\|\widetilde{r} - r\|_1 \leq \epsilon_0/4, \|\widetilde{c} - c\|_1 \leq \epsilon_0/4$ and $\min_{i \in [n]} \widetilde{r}_i \geq \epsilon_0/(8n), \min_{j \in [n]} \widetilde{c}_j \geq \epsilon_0/(8n)$.
6:     $(\widetilde{r}, \widetilde{c}) \leftarrow (1 - \frac{\epsilon_0}{8})((r, c) + \frac{\epsilon_0}{n(8-\epsilon_0)}(\mathbf{1}_n, \mathbf{1}_n))$
7:     $(u, v, L, w) \leftarrow$ SINKHORNALGORITHM($\widetilde{r}, \widetilde{c}, \epsilon_0/2$)                    ▷ Algorithm 4
8:                          ▷ Note that $u, v, L, w$ is an implicit representation of $B$, i.e., $\mathrm{diag}(e^{u_k})(L_A L_A^\top) \mathrm{diag}(e^{v_k}) + \mathrm{diag}(e^{u_k})(ww^\top)\mathrm{diag}(e^{v_k})$
9:     $(p, q, X, Y, w, u, v) \leftarrow$ ROUND($u, v, L, w, r, c$)                        ▷ Algorithm 2
10:    **return** $(p, q, X, Y, L, w, u, v)$                        ▷ We return $\widehat{X}$ in an implicit way, i.e., $\widehat{X} := XBY + pq^\top/\|p\|_1$
11: **end procedure**
---

Here we introduce the implicit form of $K$ to make use of the small treewidth setting.

**Lemma 4.1.** *We assume $C = MM^\top \in \mathbb{R}^{n \times n}$, where $M \in \mathbb{R}^{n \times d}$ has treewidth $\tau$. Given $A := K - D$, where $D_{i,j} := 1$ for $i, j \in [n]$ and $K$ is defined in Definition 3.1, the Cholesky factor $L_A$ for $A = L_A L_A^\top$ is $\tau$-sparse in columns.*

*Proof.* Given $C = MM^\top$ and $M$ has treewidth $\tau$, the Cholesky factor $L_C$ for $C = MM^\top = L_M L_M^\top$ is $\tau$ sparse in column by using Lemma 2.10. As $A_{i,j} = e^{-C_{i,j}/\gamma} - 1$, we have $A_{i,j} = 0$ when $C_{i,j} = 0$. Hence, matrix $A$ is as sparse as matrix $C$. We have that the Cholesky factor $L_A$ for $A = L_A L_A^\top$ is as sparse as $L_M$. As $L_M$ is $\tau$-sparse, we complete the proof. $\square$

## 4.2 RUNNING TIME OF SINKHORN WITH SMALL TREEWIDTH

This section is to prove the running time of Algorithm 4.

**Theorem 4.2** (Running time of Algorithm 4, informal version of Theorem C.1). *Given the cost matrix $C \in \mathbb{R}^{n \times n}$ with small treewidth $\tau$ and two simplex $r, c \in \mathbb{R}_+^n$, there is an algorithm (Algorithm 4) that takes $O(n\tau)$ for each iteration and $O(n\tau^2)$ for initialization to output such that $B(u_k, v_k) \in \mathbb{R}^{n \times n}$ can be constructed (implicitly) by*

$$B(u_k, v_k) = \mathrm{diag}(e^{u_k})(L_A L_A^\top)\,\mathrm{diag}(e^{v_k}) + \mathrm{diag}(e^{u_k})(ww^\top)\,\mathrm{diag}(e^{v_k}),$$

*satisfying $\|B(u_k, v_k)\mathbf{1}_n - r\|_1 + \|B(u_k, v_k)^\top \mathbf{1}_n - c\|_1 \le \epsilon_0$.*

## 4.3 CORRECTNESS OF ROUNDING ALGORITHM

We first show the correctness of our rounding algorithm (Algorithm 2).

**Lemma 4.3** (Informal version of Lemma A.5). *Given $r, c \in \triangle_n$, $B \in \mathbb{R}_+^{n \times n}$, $u, v \in \mathbb{R}^n$ and $r, c \in \mathbb{R}^n$, there is an algorithm (Algorithm 2) that outputs a diagonal matrix $X \in \mathbb{R}^{n \times n}$, a diagonal matrix $Y \in \mathbb{R}^{n \times n}$, a lower triangular matrix $L_A$, vectors $u, v, w \in \mathbb{R}^n$, vectors $p \in \mathbb{R}^n$, $q \in \mathbb{R}^n$. Such that $G \in \mathcal{U}_{r,c}$ can be constructed (implicitly) by*

$$\widehat{X} = 7\,X(\mathrm{diag}(e^u)L_A L_A^\top \mathrm{diag}(e^v) + \mathrm{diag}(e^u)(ww^\top)\mathrm{diag}(e^v))Y + pq^\top / \|p\|_1,$$

*satisfying $\|G - B\|_1 \le 2(\|B\mathbf{1}_n - r\|_1 + \|B^\top \mathbf{1}_n - c\|_1)$.*

---

**Algorithm 2** Rounding of the projection of $B$ on $\mathcal{U}$

---

1: **procedure** ROUND($u \in \mathbb{R}^n, v \in \mathbb{R}^n, L \in \mathbb{R}^{n \times n}, w \in \mathbb{R}^n, r \in \mathbb{R}^n, c \in \mathbb{R}^n$)  $\triangleright$ Lemma 4.4
2:  $\qquad\qquad\qquad\qquad\qquad\quad$ $\triangleright$ $L$ is a lower triangular matrix that only has $O(n\tau)$ nonzeros
3:  $\qquad\qquad\qquad\qquad\quad$ $\triangleright$ We never explicit write $B$. $B$ can implicitly represented by
  $\mathrm{diag}(e^u)(L_A L_A^\top)\,\mathrm{diag}(e^v) + \mathrm{diag}(e^u)(ww^\top)\,\mathrm{diag}(e^v)$
4:  $\quad X \leftarrow \mathrm{diag}(x)$ with $x_i = \min\{\frac{r_i}{r_i(B)}, 1\}$  $\qquad\qquad$ $\triangleright$ $r(B) := B\mathbf{1}_n, X \in \mathbb{R}^{n \times n}$
5:  $\quad B_0 \leftarrow XB$  $\qquad\qquad\qquad\qquad\qquad\quad$ $\triangleright$ We only implicitly construct $B_0$
6:  $\quad Y \leftarrow \mathrm{diag}(y)$ with $y_j = \min\{\frac{c_j}{c_j(B_0)}, 1\}$  $\qquad\qquad\quad$ $\triangleright$ $c(B_0) := B_0^\top \mathbf{1}_n$
7:  $\quad B_1 \leftarrow B_0 Y$  $\qquad\qquad\qquad\qquad\qquad\quad$ $\triangleright$ We only implicitly construct $B_1$
8:  $\quad p \leftarrow r - B_1 \mathbf{1}_n$
9:  $\quad q \leftarrow c - B_1^\top \mathbf{1}_n$
10:  $\quad$ **return** $p, q, X, Y, w, u, v$  $\triangleright$ We return $G$ in an implicit way, i.e., $G := XBY + pq^\top / \|p\|_1$
11: **end procedure**

---

## 4.4 RUNNING TIME OF ROUNDING ALGORITHM

Next, we show the running time needed for the rounding algorithm (Algorithm 2).

**Lemma 4.4** (An improved version of of Lemma 7 in (Altschuler et al., 2017)). *Given $r, c \in \triangle_n$, $B \in \mathbb{R}_+^{n \times n}$, $u, v \in \mathbb{R}^n$ and $r, c \in \mathbb{R}^n$, there is an algorithm (Algorithm 2) that outputs a diagonal matrix $X \in \mathbb{R}^{n \times n}$, a diagonal matrix $Y \in \mathbb{R}^{n \times n}$, a lower triangular matrix $L_A$, vectors $u, v, w \in \mathbb{R}^n$, vectors $p \in \mathbb{R}^n$, $q \in \mathbb{R}^n$, such that $G \in \mathcal{U}_{r,c}$ can be constructed (implicitly) by*

$$\widehat{X} = X(\mathrm{diag}(e^u)L_A L_A^\top \mathrm{diag}(e^v) + \mathrm{diag}(e^u)(ww^\top)\mathrm{diag}(e^v))Y + pq^\top / \|p\|_1,$$

*satisfying $\|G - B\|_1 \le 2(\|B\mathbf{1}_n - r\|_1 + \|B^\top \mathbf{1}_n - c\|_1)$ in $O(n\tau)$ time.*

*Proof.* The running time for each step is shown as follows:

- Calculating $r(B)$ takes $O(n\tau)$ time. Given

$$r(B) = B\mathbf{1}_n = \mathrm{diag}(e^{u_k})(L_A L_A^\top)\mathbf{1}_n \mathrm{diag}(e^{v_k}) + \mathrm{diag}(e^{u_k})(ww^\top)\mathbf{1}_n \mathrm{diag}(e^{v_k}),$$

  calculating $L_A(L_A^\top \mathbf{1}_n)$ takes $O(n\tau)$, as $\mathrm{nnz}(L_A) = n\tau$. As $w = \mathbf{1}_n$, calculating $(ww^\top)\mathbf{1}_n$ takes $O(n)$.

- Calculating $X = \mathrm{diag}(x)$ with $x_i = \min\{\frac{r_i}{r_i(B)}, 1\}$ takes $O(n)$ time.

- For $B_0 = XB$, we remark that $B_0$ is not explicitly written down. It is implicitly represented by $L_A, w, u, v, X$.

- Similarly, we can calculate $Y$ in $O(n)$ time and implicitly write down $B_1$.

- We have

$$B_1 \mathbf{1}_n = XBY$$
$$= \mathrm{diag}(e^{u_k})X(L_A L_A^\top)Y\mathbf{1}_n \mathrm{diag}(e^{v_k}) + \mathrm{diag}(e^{u_k})X(ww^\top)Y\mathbf{1}_n \mathrm{diag}(e^{v_k}).$$

  For any diagonal matrix $M$, $M \cdot L_A$ is as sparse as $L_A$ and it takes $O(n\tau)$ to compute it. Therefore, computing $P = \mathrm{diag}(e^{u_k})X(L_A L_A^\top)Y \mathrm{diag}(e^{v_k})$ takes $O(n\tau)$ time and $P$ is $n\tau$-sparse. Then, we compute $P\mathbf{1}_n$, which takes $O(n\tau)$ time. Hence, updating $p$ takes $O(n\tau)$ time.

- Similarly, updating $q$ takes $O(n\tau)$ time.

- For matrix $G$, it is returned in an implicit way. We use $p, q, X, Y, w, u, v$ to represent it.

Therefore, the total running time is $O(n\tau)$. $\qquad\square$

## 4.5 Running time of OT Distance by Sinkhorn

Putting them all together, we have the following theorem:

**Theorem 4.5** (Informal version of Theorem B.3). *There is an algorithm (Algorithm 1) that takes cost matrix $C \in \mathbb{R}^{n \times n}$, two $n$-dimensional simplex $r, c$ as inputs and outputs, a diagonal matrix $X \in \mathbb{R}^{n \times n}$, a diagonal matrix $Y \in \mathbb{R}^{n \times n}$, a lower triangular matrix $L_A$, vectors $u, v, w \in \mathbb{R}^n$, vectors $p \in \mathbb{R}^n$, $q \in \mathbb{R}^n$. Such that $\widehat{X} \in \mathcal{U}(r, c)$ can be constructed (implicitly) by*

$$\widehat{X} = X(\mathrm{diag}(e^u)L_A L_A^\top \mathrm{diag}(e^v) + \mathrm{diag}(e^u)(ww^\top)\mathrm{diag}(e^v))Y + pq^\top/\|p\|_1$$

*that satisfying Eq. (2) in*

$$O(n\tau^2 + \epsilon^{-2}n\tau\|C\|_\infty^2 \ln n)$$

*time.*

## 5 Conclusion

Optimal transport (OT) such as the earth mover's distance is a specialized domain within mathematics that delves into the intricate problem of determining the most economical way to transport goods or materials between two distinct points. These costs are typically quantified based on parameters such as the total distance traversed or the amount of resources utilized during the transportation process. In this paper, we study the problem of approximating the general OT distance between two discrete distributions of size $n$. Consider the cost matrix denoted by $C = AA^\top$ where $A \in \mathbb{R}^{n \times d}$ with treewidth $\tau$. By making use of the treewidth setting, we proposed a faster Sinkhorn algorithm and a faster rounding algorithm to approximate the OT distance. The existing state-of-the-art algorithms have running time of $\widetilde{O}(\epsilon^{-2}n^2)$. In contrast, our approach has improve this time to $\widetilde{O}(\epsilon^{-2}n\tau)$.

## ETHIC STATEMENT

This paper does not involve human subjects, personally identifiable data, or sensitive applications. We do not foresee direct ethical risks. We follow the ICLR Code of Ethics and affirm that all aspects of this research comply with the principles of fairness, transparency, and integrity.

## REPRODUCIBILITY STATEMENT

We ensure reproducibility of our theoretical results by including all formal assumptions, definitions, and complete proofs in the appendix. The main text states each theorem clearly and refers to the detailed proofs. No external data or software is required.

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

# Appendix

**Roadmap** We first provide the formal analysis of the Sinkhorn's algorithm in Section A. Then, we show the detailed running time proof in Section B. In Section C, we provide the missing proof and algorithm. In addition, in Section D, we provide the analysis of the Sinkhorn's algorithm in symmetric case.

## A MORE ANALYSIS FOR SINKHORN'S ALGORITHM

In Section A.1, we introduce the Sinkhorn's algorithm. In Section A.2, we provide a lemma that guarantee the bounds related to $u \in \mathbb{R}^n, v \in \mathbb{R}^n$. In Section A.3, we show the iteration complexity bound of Sinkhorn's algorithm. In Section A.4, we show the correctness of our rounding algorithm (Algorithm 2) under the treewidth setting.

### A.1 SINKHORN ALGORITHM

We first provide the Sinkhorn algorithm (Algorithm 3) as follows:

---
**Algorithm 3** Sinkhorn's Algorithm
---
1: **procedure** SINKHORNALGORITHM($c, r, \epsilon_0$)              ▷ Lemma A.4
2:                                                              ▷ Accuracy $\epsilon_0$
3:      $k \leftarrow 0$
4:      $u_0 \leftarrow 0$
5:      $v_0 \leftarrow 0$
6:      **while** $\|B(u_k, v_k)\mathbf{1}_n - r\|_1 + \|B(u_k, v_k)^\top \mathbf{1}_n - c\|_1 \geq \epsilon_0$ **do**
7:          **if** $k \mod 2 = 0$ **then**
8:              $u_{k+1} \leftarrow u_k + \ln r - \ln(B(u_k, v_k)\mathbf{1}_n)$
9:              $v_{k+1} \leftarrow v_k$
10:          **else**
11:              $v_{k+1} \leftarrow v_k + \ln c - \ln(B(u_k, v_k)^\top \mathbf{1}_n)$
12:              $u_{k+1} \leftarrow u_k$
13:          **end if**
14:          $k \leftarrow k + 1$
15:      **end while**
16:      **return** $B(u_k, v_k)$.
17: **end procedure**

---

### A.2 BOUNDED $\max - \min$

Next, we provide a tool related to the bound for $u_k$ and $v_k$.

**Lemma A.1.** *Let $k \geq 0$ and $u_k \in \mathbb{R}^n, v_k \in \mathbb{R}^n$ be generated by Algorithm 3 and $(u_*, v_*) \in \mathbb{R}^n \times \mathbb{R}^n$ be a solution of Eq.* (5). *Then*

$$\max_{i \in [n]} u_{k,i} - \min_{i \in [n]} u_{k,i} \leq R, \ \max_{j \in [n]} v_{k,j} - \min_{j \in [n]} v_{k,j} \leq R, \tag{8}$$

$$\max_{i \in [n]} u_{*,i} - \min_{i \in [n]} u_{*,i} \leq R, \ \max_{j \in [n]} v_{*,j} - \min_{j \in [n]} v_{*,j} \leq R,$$

*where $R$ is defined in Definition 3.3.*

*Proof.* First, we prove the bound for $u_k \in \mathbb{R}^n$. As $u, v$ are initialized as $\mathbf{0}_n$, the inequality holds for $k = 0$. Given $k - 1$ is even, the variable $u$ is updated on the iteration $k - 1$ and $B(u_k, v_k)\mathbf{1}_n = r$ by the algorithm construction.

Hence, for each $i \in [n]$, we have

$$e^{u_{k,i}} K_{\min} \langle \mathbf{1}_n, e^{v_k} \rangle \leq \sum_{j=1}^n e^{e_{k,i}} K_{i,j} e^{v_{k,j}}$$

$$= [B(u_k, v_k)(\mathbf{1}_n)_i]$$
$$= r_i$$
$$\leq 1, \tag{9}$$

where the first step follows from the definition of $K_{\min}$, the second step follows from the definition of $B$, the third step follows from $B(u_k, v_k)\mathbf{1}_n = r$ and the last step follows from the definition of probability simplex $r$.

Hence, by reorganizing Eq. (9) we have

$$\max_{i \in [n]} u_{k,i} \leq -\ln(K_{\min}\langle \mathbf{1}_n, e^{v_k}\rangle). \tag{10}$$

On the other hand, since $0 \leq K_{i,j} \leq 1$ for each $i \in [n]$,

$$e^{u_{k,i}}\langle \mathbf{1}_n, e^{v_k}\rangle$$
$$\geq \sum_{j=1}^{n} e^{u_{k,i}} K_{i,j} e^{v_{k,j}}$$
$$= [B(u_k, v_k)\mathbf{1}_n]_i$$
$$= r_i,$$

where the first step follows from $K_{i,j} \leq 1$, the second step follows from the definition of $B$ and the last step follows from $B(u_k, v_k)\mathbf{1}_n = r$.

We also have

$$\min_{i \in [n]} u_{k,i} \geq \min_{i \in [n]} \ln\left(\frac{r_i}{\langle \mathbf{1}_n, e^{v_k}\rangle}\right) = \ln\left(\frac{\min_{i \in [n]} r_i}{\langle \mathbf{1}_n, e^{v_k}\rangle}\right).$$

The latter equality and Eq. (10) give

$$\max_{i \in [n]} u_{k,i} - \min_{i \in [n]} u_{k,i} \leq -\ln(K_{\min} \min_{i \in [n]} r_i) \leq R.$$

$\square$

### A.3 ITERATION COMPLEXITY BOUND

Then, we show that the upper bound of the potential function $\widetilde{\psi}$.

**Lemma A.2** (Formal version of Lemma 3.6). *Let $k \geq 1$ and $u_k, v_k \in \mathbb{R}^n$ be output of Algorithm 3. We denote $B_k := B(u_k, v_k)$. Then, we have*

$$\widetilde{\psi}(u_k, v_k) \leq R \cdot (\|B_k\mathbf{1}_n - r\|_1 + \|B_k^\top\mathbf{1}_n - c\|_1).$$

*Proof.* Given a fixed $k \geq 1$, for the following convex function of $(\widehat{u}, \widehat{v})$

$$\langle \mathbf{1}_n, B(\widehat{u}, \widehat{v})\mathbf{1}_n\rangle - \langle \widehat{u}, B(u_k, v_k)\mathbf{1}_n\rangle - \langle \widehat{v}, B(u_k, v_k)^\top\mathbf{1}_n\rangle.$$

The gradient of the convex function vanishes at $(\widehat{u}, \widehat{v}) = (u_k, v_k)$, so the point $(u_k, v_k)$ is its minimizer.

Hence,

$$\widetilde{\psi}(u_k, v_k) = [\langle \mathbf{1}_n, B_k\mathbf{1}_n\rangle - \langle u_k, B_k\mathbf{1}_n\rangle - \langle v_k, B_k^\top\mathbf{1}_n\rangle]$$
$$- [\langle \mathbf{1}_n, B(u_*, v_*)\mathbf{1}_n\rangle$$
$$- \langle u_*, B_k\mathbf{1}_n\rangle - \langle v_*, B_k^\top\mathbf{1}_n\rangle]$$
$$+ \langle u_k - u_*, B_k\mathbf{1}_n - r\rangle$$
$$+ \langle v_k - v_*, B_k^\top\mathbf{1}_n - c\rangle$$
$$\leq \langle u_k - u_*, B_k\mathbf{1}_n - r\rangle$$
$$+ \langle v_k - v_*, B_k^\top\mathbf{1}_n - c\rangle. \tag{11}$$

where the first step follows from the definition of $\widetilde{\psi}$. Next, we bound the r.h.s of the inequality. For each iteration, we know that either $B_k \mathbf{1}_n = r$ or $B_k^\top \mathbf{1}_n = c$, so we have that $\langle \mathbf{1}_n, B_k \mathbf{1}_n \rangle = 1$ and $\langle \mathbf{1}_n, B_k \mathbf{1}_n - r \rangle = 0$.

Taking $a = 0.5 \cdot (\max_{i \in [n]} u_{k,i} + \min_{i \in [n]} u_{k,i})$. Then, we have

$$
\begin{aligned}
\langle u_k, B_k \mathbf{1}_n - r \rangle &= \langle u_k - a\mathbf{1}_n, B_k \mathbf{1}_n - r \rangle \\
&\leq \|u_k - a\mathbf{1}_n\|_\infty \|B_k \mathbf{1}_n - r\|_1 \\
&= 0.5 \cdot (\max_{i \in [n]} u_{k,i} \\
&\qquad - \min_{i \in [n]} u_{k,i}) \|B_k \mathbf{1}_n - r\|_1 \\
&\leq \frac{R}{2} \|B_k \mathbf{1}_n - r\|_1.
\end{aligned}
$$

where the first step follows from $\langle \mathbf{1}_n, B_k \mathbf{1}_n - r \rangle = 0$, the second step follows from Hölder's inequality, the third step follows from the definition of $a$, and the last step follows from Lemma A.1.

Similarly, we bound $\langle -u_*, B_k \mathbf{1}_n - r \rangle$, $\langle v_k, B_k^\top \mathbf{1}_n - c \rangle$ and $\langle -v_*, B_k^\top \mathbf{1}_n - c \rangle$ in Eq. (11) and complete the proof. □

The following lemma gives the lower bound of $\widetilde{\psi}(u_k, v_k) - \widetilde{\psi}(u_{k+1}, v_{k+1})$.

**Lemma A.3.** *Let $u_k, v_k$ be as in Algorithm 3. Then,*

$$
\widetilde{\psi}(u_k, v_k) - \widetilde{\psi}(u_{k+1}, v_{k+1}) \geq \max\{\frac{\widetilde{\psi}(u_k, v_k)^2}{2R^2}, \frac{\epsilon_0^2}{2}\}.
$$

*Proof.* We first consider case when $k \geq 1$ is even and define $B_k := B(u_k, v_k)$, where $B$ is as in Definition 3.1. We have

$$
\begin{aligned}
\psi(u_k, v_k) - \psi(u_{k+1}, v_{k+1}) &= \langle \mathbf{1}_n, B_k \mathbf{1}_n \rangle - \langle \mathbf{1}_n, B_{k+1} \mathbf{1}_n \rangle + \langle u_{k+1} - u_k, r \rangle + \langle v_{k+1} - v_k, c \rangle \\
&= \langle r, u_{k+1} - u_k \rangle \\
&= \langle r, \ln r - \ln(B_k \mathbf{1}_n) \rangle \\
&= \mathrm{KL}(r \| B_k \mathbf{1}_n). \qquad (12)
\end{aligned}
$$

Then, we obtain

$$
\begin{aligned}
\widetilde{\psi}(u_k, v_k) - \widetilde{\psi}(u_{k+1}, v_{k+1}) &= \psi(u_k, v_k) - \psi(u_{k+1}, v_{k+1}) \\
&= \mathrm{KL}(r \| B_k \mathbf{1}_n) \\
&\geq \frac{1}{2} \|B_k \mathbf{1}_n - r\|_1^2 \\
&\geq \max\{\frac{\widetilde{\psi}(u_k, v_k)^2}{2R^2}, \frac{\epsilon_0^2}{2}\},
\end{aligned}
$$

where the first step follows by the definition of $\widetilde{\psi}$, the second step follows by Eq. (12), the third step follows by Pinsker's inequality and the last step follows by Lemma 3.6 and $B_k^\top \mathbf{1}_n = c$. For the last step, we also used that, as soon as the stopping criterion is not yet fulfilled and $B_k^\top \mathbf{1}_n = c$, $\|B_k \mathbf{1}_n - r\|_1^2 \geq \epsilon_0^2$. Similarly, when $k$ is odd, we can prove the same inequality.

□

Here, we show the upper bound of the number of iterations required for Algorithm 3.

**Lemma A.4.** *Given the cost matrix $C \in \mathbb{R}^{n \times n}$ and two simplex $r, c \in \mathbb{R}_+^n$, there is an algorithm (Algorithm 3) outputs $B(u_k, v_k)$ (Definition 3.1) that satisfying*

$$
\|B(u_k, v_k) \mathbf{1}_n - r\|_1 + \|B(u_k, v_k)^\top \mathbf{1}_n - c\|_1 \leq \epsilon_0
$$

*in the number of iterations $k$ satisfying*

$$
k \leq 2 + \frac{4R}{\epsilon_0}
$$

*Proof.* Given $\ell = \frac{2R^2}{\widetilde{\psi}(u_1,v_1)}$ in accordance with Lemma 3.7, we have for any $k \geq 1$

$$\frac{\widetilde{\psi}(u_k,v_k)}{2R^2} \leq \frac{1}{k+\ell-1}.$$

Thus,

$$k \leq 1 + \frac{2R^2}{\widetilde{\psi}(u_k,v_k)} - \frac{2R^2}{\widetilde{\psi}(u_1,v_1)}. \tag{13}$$

On the other hand,

$$\widetilde{\psi}(u_{k+m},v_{k+m}) \leq \widetilde{\psi}(u_k,v_k) - \frac{\epsilon_0^2 m}{2}, \quad k,m \geq 0. \tag{14}$$

Next, we use a switching strategy, parameterized by number $s \in (0, \widetilde{\psi}(u_1,v_1)]$, to combine Eq. (7) and Eq. (14).

First, by using Eq. (7), we calculate the number of iterations needed to decrease $\widetilde{\psi}(u,v)$ from its initial value $\widetilde{\psi}(u_1,v_1)$ to a certain value $s$. Then, by applying Eq. (14) and given $\widetilde{\psi}(u,v) \geq 0$ by its definition, we calculate the number of iterations required to further decrease $\widetilde{\psi}(u,v)$ from $s$ to zero. By minimizing the sum of these two estimates in $s \in (0, \widetilde{\psi}(u_1,v_1)]$, the total number of iterations $k$ satisfies the following

$$k \leq \min_{0 < s \leq \widetilde{\psi}(u_1,v_1)} \left(2 + \frac{2R^2}{s} - \frac{2R^2}{\widetilde{\psi}(u_1,v_1)} + \frac{2s}{\epsilon_0^2}\right)$$

$$= \begin{cases} 2 + \frac{4R}{\epsilon_0} - \frac{2R^2}{\widetilde{\psi}(u_1,v_1)}, & \widetilde{\psi}(u_1,v_1) \geq R\epsilon_0, \\ 2 + \frac{2\widetilde{\psi}(u_1,v_1)}{\epsilon_0^2}, & \widetilde{\psi}(u_1,v_1) < R\epsilon_0. \end{cases}$$

where the first step comes from Eq. (13), the first half of the last step comes from $a + b \geq 2\sqrt{ab}$ for $a \geq 0$, $b \geq 0$ and the second half of the last step follows from $s = \widetilde{\psi}(u_1,v_1)$. In both cases, we have $k \leq 2 + \frac{4R}{\epsilon_0}$. $\qquad\square$

### A.4 CORRECTNESS OF ROUNDING ALGORITHM

We show the correctness of our rounding algorithm (Algorithm 2).

**Lemma A.5** (Formal version of Lemma 4.3. An improved version of of Lemma 7 in (Altschuler et al., 2017)). *Given $r, c \in \triangle_n$, $B \in \mathbb{R}_+^{n \times n}$, $u, v \in \mathbb{R}^n$ and $r, c \in \mathbb{R}^n$, there is an algorithm (Algorithm 2) that outputs*

- *a diagonal matrix $X \in \mathbb{R}^{n \times n}$*

- *a diagonal matrix $Y \in \mathbb{R}^{n \times n}$*

- *a lower triangular matrix $L_A$*

- *vectors $u, v, w \in \mathbb{R}^n$*

- *vectors $p \in \mathbb{R}^n$, $q \in \mathbb{R}^n$*

*such that $G \in \mathcal{U}_{r,c}$ can be constructed (implicitly) by*

$$\widehat{X} = X(\mathrm{diag}(e^u)L_A L_A^\top \mathrm{diag}(e^v) + \mathrm{diag}(e^u)(ww^\top)\mathrm{diag}(e^v))Y$$
$$+ pq^\top/\|p\|_1,$$

*satisfying*

$$\|G - B\|_1 \leq 2(\|B\mathbf{1}_n - r\|_1 + \|B^\top \mathbf{1}_n - c\|_1).$$

*Proof.* Let $G$ be the output of Algorithm 2. As matrix $B_1$ are nonnegative, and the output $q$ and $p$ are both negative, with $\|p\|_1 = \|q\|_1 = 1 - \|B_1\|_1$, matrix $G$ are nonnegative and

$$r(G) = r(B_1) + r(pq^\top/\|p\|_1)$$
$$= r(B_1) + p$$
$$= r, \tag{15}$$

where we denote $r(A) := A\mathbf{1}_n, c(A) := A^\top\mathbf{1}_n$ and the first two step comes from the definition of $r$ and the last step comes from $p = r - B_1\mathbf{1}_n$. Similarly, we have $c(G) = c$. Therefore, we have $G \in \mathcal{U}_{r,c}$.

Next, we denote $\Delta := \|B\|_1 - \|B_1\|_1$ and prove the $\ell_1$ bound between the matrix $B$ and matrix $G$. We first remove mass from a row of $B$ when $r_i(B) \geq r_i$, and then, we remove mass from a column when $c_j(B_0) \geq c_j$. Now, we have

$$\Delta = \sum_{i=1}^{n}(r_i(B) - r_i)_+ + \sum_{j=1}^{n}(c_j(B_0) - c_j)_+. \tag{16}$$

Then, we show the analysis of Eq. (16). First, for the left sum of Eq. (16), we have

$$\sum_{i=1}^{n}(r_i(B) - r_i)_+ = \frac{1}{2}(\|r(B) - r\|_1 + \|B\| - 1).$$

For the second sum in Eq. (16).

$$\sum_{j=1}^{n}(c_j(B_0) - c_j)_+ \leq \sum_{j=1}^{n}(c_j(B) - c_j)_+ \leq \|c(B) - c\|_1,$$

where the first step comes from the fact that the vector $c(B)$ is entrywise larger than $c(B_0)$ and the last step comes from the definition of $c$.

Therefore we conclude

$$\|G - B\|_1 \leq \Delta + \|pq^\top\|_1/\|p\|_1$$
$$= \Delta + 1 - \|B_1\|_1$$
$$= 2\Delta + 1 - \|B\|_1$$
$$\leq \|r(B) - r\|_1 + 2\|c(B) - c\|_1 \tag{17}$$
$$\leq 2(\|r(B) - r\|_1 + \|c(B) - c\|_1),$$

where the first step comes from the definition of $\Delta$, the second step comes from the fact that $\|p\|_1 = \|q\|_1 = 1 - \|B_1\|_1$, the third step comes from the definition of $\Delta$, the fourth step comes from Eq. (16) and the last step comes from reorganization. Now we complete the proof. $\square$

## B    RUNNING TIME

In Section B.1, we present two inequalities. In Section B.2, we show the running time needed for our rounding algorithm (Algorithm 2) under the treewidth setting. In Section B.3, we show the running time needed for approximating the OT by Sinkhorn (Algorithm 1).

### B.1    INEQUALITIES

We introduce the Hölder's inequality as following:

**Lemma B.1** (Hölder's inequality). *If $p > 1$ and $q > 1$ are such that*

$$\frac{1}{p} + \frac{1}{q} = 1,$$

*then*

$$\|ab\|_1 \leq \|a\|_p\|b\|_q.$$

We also provide the Pinsker inequality.

**Lemma B.2** (Pinsker inequality). *Let $P$ and $Q$ be two distributions defined on the universe $U$. Then,*

$$\mathrm{KL}(P\|Q) \geq \frac{1}{2\ln 2} \cdot \|P - Q\|_1^2.$$

*where $\mathrm{KL}(P\|Q)$ is the $\mathrm{KL}$-divergence between $P$ and $Q$.*

### B.2 Running Time of Rounding Algorithm

We show the running time needed for the rounding algorithm (Algorithm 2).

### B.3 Running Time of OT Distance by Sinkhorn

The core of our OT algorithm is the entropic penalty

$$X_\gamma := \arg\min_{X \in \mathcal{U}_{r,c}} \langle X, C \rangle + \gamma \cdot \mathcal{R}(X). \tag{18}$$

The solution to Eq. (18) can be characterized explicitly by analyzing its first-order conditions for optimality.

Now we apply the result of the previous subsection to derive a complexity estimate for finding $\widehat{X} \in \mathcal{U}(r, c)$ satisfying Eq. (2). The procedure for approximating the OT distance by the Sinkhorn's algorithm is listed as Algorithm 1.

**Theorem B.3** (Formal version of Theorem 4.5). *There is an algorithm (Algorithm 1) that takes cost matrix $C \in \mathbb{R}^{n \times n}$, two $n$-dimensional simplex $r, c$ as inputs and outputs*

- *a diagonal matrix $X \in \mathbb{R}^{n \times n}$*

- *a diagonal matrix $Y \in \mathbb{R}^{n \times n}$*

- *a lower triangular matrix $L_A$*

- *vectors $u, v, w \in \mathbb{R}^n$*

- *vectors $p \in \mathbb{R}^n$, $q \in \mathbb{R}^n$*

*such that $\widehat{X} \in \mathcal{U}(r, c)$ can be constructed (implicitly) by*

$$\widehat{X} = X(\mathrm{diag}(e^u)L_A L_A^\top \mathrm{diag}(e^v) + \mathrm{diag}(e^u)(ww^\top)\mathrm{diag}(e^v))Y + pq^\top/\|p\|_1,$$

*which satisfies Eq. (2) in*

$$O(n\tau^2 + \epsilon^{-2}n\tau\|C\|_\infty^2 \ln n)$$

*time.*

**Remark B.4.** *If we don't care about the output format to be lower-triangular matrix, then the additive term $n\tau^2$ can be removed.*

*Proof.* Let $X_* \in \arg\min_{X \in \mathcal{U}_{r,c}} \langle P, C \rangle$ be an optimal solution to the original OT program.

We first show that $\langle B, C \rangle$ is not much larger than $\langle X_*, C \rangle$.

Since $B = MAN \in \mathbb{R}^{n \times n}$ for positive diagonal matrices $M, N \in \mathbb{R}_+^{n \times n}$, Lemma 2.6 implies $B$ is the optimal solution to

$$\arg\min_{X \in \mathcal{U}_{r,c}} \langle X, C \rangle + \gamma \mathcal{R}(X). \tag{19}$$

By Lemma 4.4, there exists a matrix $X_0 \in \mathcal{U}_{B\mathbf{1}_n, B^\top \mathbf{1}_n}$ (Definition 2.5) such that

$$\|X_0 - X_*\|_1 \leq 2(\|B\mathbf{1}_n - r\|_1 + \|B^\top \mathbf{1}_n - c\|_1). \tag{20}$$

Moreover, since $B \in \mathbb{R}^{n \times n}$ is an optimal solution of Eq. (19), we have

$$\langle B, C \rangle + \gamma \mathcal{R}(B) \leq \langle X_0, C \rangle + \gamma \mathcal{R}(X_0). \tag{21}$$

Thus, we have

$$
\begin{aligned}
&\langle B, C\rangle - \langle X_*, C\rangle \\
&= \langle B, C\rangle - \langle X_0, C\rangle + \langle X_0, C\rangle - \langle X_*, C\rangle \\
&= \langle B, C\rangle - \langle X_0, C\rangle + \|X_0 - X_*\|_1 \|C\|_\infty \\
&\leq \gamma(H(B) - H(X_0)) + \|X_0 - X_*\|_1 \|C\|_\infty \\
&\leq \gamma(H(B) - H(X_0)) + 2(\|B\mathbf{1}_n - r\|_1 + \|B^\top \mathbf{1}_n - c\|_1)\|C\|_\infty \\
&\leq 2\gamma \ln n + 2(\|B\mathbf{1}_n - r\|_1 + \|B^\top \mathbf{1}_n - c\|_1)\|C\|_\infty,
\end{aligned}
\tag{22}
$$

where the first step follows from reorganization, the second step follows from Hölder's inequality (Lemma B.1), the third step follows from Eq. (21) and $\mathcal{R}(X) = -H(X)$, the fourth step follows from Eq. (20) and the last step follows from the fact that $0 < H(B), H(X_0) \leq 2\ln n$.

Lemma 4.4 implies that the output $\widehat{X}$ of Algorithm 2 satisfies the inequality

$$
\|B - \widehat{X}\|_1 \leq 2(\|B\mathbf{1}_n - r\|_1 + \|B^\top \mathbf{1}_n - c\|_1).
\tag{23}
$$

Recall that $\widehat{X}$ is the output of Algorithm 1, $X_*$ is a solution to the OT problem Eq. (2) and $B$ is the matrix obtained in line 7 of Algorithm 1. We have

$$
\begin{aligned}
\langle \widehat{X}, C\rangle &= \langle \widehat{X} - B, C\rangle + \langle B, C\rangle \\
&\leq \|\widehat{X} - B\|_1 \|C\|_\infty + \langle B, C\rangle \\
&\leq 2(\|B\mathbf{1}_n - r\|_1 + \|B^\top \mathbf{1}_n - c\|_1)\|C\|_\infty + \langle B, C\rangle \\
&\leq \langle X_*, C\rangle + 2\gamma \ln n + 4(\|B\mathbf{1}_n - r\|_1 + \|B^\top \mathbf{1}_n - c\|_1)\|C\|_\infty.
\end{aligned}
\tag{24}
$$

where the first step follows from reorganization, the second step follows from Hölder's inequality, the third step follows from Eq. (23) and the last step follows from Eq. (22).

At the same time, we have

$$
\begin{aligned}
&\|B\mathbf{1}_n - r\|_1 + \|B^\top \mathbf{1}_n - c\|_1 \\
&\leq \|B\mathbf{1}_n - \widetilde{r}\|_1 + \|\widetilde{r} - r\|_1 + \|B^\top \mathbf{1}_n - \widetilde{c}\|_1 + \|\widetilde{c} - c\|_1 \\
&\leq \epsilon_0,
\end{aligned}
$$

where the first step follows from the definition of $\ell_1$-norm and the last step follows from $\|B\mathbf{1}_n - r\|_1 + \|B^\top \mathbf{1}_n - c\|_1 \leq \epsilon_0$ (output of Algorithm 4) and the definitions of $\widetilde{r}$ and $\widetilde{c}$.

Setting $\gamma = \frac{\epsilon}{4\ln n}$ and $\epsilon_0 = \frac{\epsilon}{8\|C\|_\infty}$, we obtain from the above inequality and Eq. (24) that $\widehat{X}$ satisfies inequality Eq. (2).

Next, we show complexity of Algorithm 1. When $\epsilon_0$ is sufficiently small, the number of iterations of the Sinkhorn's algorithm in line 7 of Algorithm 1 is $O(R/\epsilon_0)$, by using Theorem A.4. According to Definition 3.3, we have

$$
\begin{aligned}
R &= -\ln(K_{\min} \min_{i,j\in[n]} \{\widetilde{r}_i, \widetilde{c}_j\}) \\
&= -\ln(e^{-\|C\|_\infty/\gamma} \min_{i,j\in[n]} \{\widetilde{r}_i, \widetilde{c}_j\}) \\
&\leq \frac{\|C\|_\infty}{\gamma} - \ln(\frac{\epsilon_0}{8n}),
\end{aligned}
$$

where the first step follows from the definition of $R$, the second step follows from the definition of $K_{\min}$, the last step follows from the condition of $\widetilde{r}_i, \widetilde{c}_j$ in line 6 of Algorithm 1.

Since $\gamma = \frac{\epsilon}{4\ln n}$ and $\epsilon_0 = \frac{\epsilon}{8\|C\|_\infty}$, we have that

$$
R = O(\epsilon^{-1}\|C\|_\infty \ln n).
$$

As the number of iteration for Algorithm 1 is $O(R/\epsilon_0)$, we conclude that the total number of Sinkhorn's algorithm iterations is bounded by $O(\epsilon^{-2}\|C\|_\infty^2 \ln n)$.

Obviously, $\widetilde{r} \in \mathbb{R}^n_+$ and $\widetilde{c} \in \mathbb{R}^n_+$ in line 6 of Algorithm 1 can be found in $O(n)$ time.

Since each iteration of the Sinkhorn's algorithm requires $O(n\tau)$ time and the initialization takes $O(n\tau^2)$ time as shown in Theorem 4.2, the total complexity of Algorithm 1 is

$$O(n\tau^2 + \epsilon^{-2} n\tau \|C\|^2_\infty \ln n).$$

$\square$

## C  SINKHORN ALGORITHM WITH SMALL TREEWIDTH

---

**Algorithm 4** Sinkhorn's Algorithm with small treewidth

---

1: **procedure** SINKHORNALGORITHM($r \in \mathbb{R}^n, c \in \mathbb{R}^n, \epsilon_0 \in (0,1)$)      ▷ Theorem 4.2
2:                                                                       ▷ Accuracy $\epsilon_0$
3:        $k \leftarrow 0,$
4:        $u_0 \leftarrow 0$
5:        $v_0 \leftarrow 0$
6:        $w \leftarrow \mathbf{1}_n$
7:        $x_0 \leftarrow e^{-C_{i,j}/\gamma} \mathbf{1}_n$
8:        $y_0 \leftarrow (e^{-C_{i,j}/\gamma})^\top \mathbf{1}_n$
9:        Implicitly form $D = ww^\top$
10:       Implicitly form $A \in \mathbb{R}^{n \times n}$, where $A_{i,j} = e^{-C_{i,j}/\gamma} - 1$
11:          ▷ Explicitly writing down $A$ requires $n^2$, however, we never need to explicitly write down $A$. Knowing the exact formulation of $A$ is enough to do the Cholesky decomposition
12:       $L \leftarrow$ Cholesky decomposition matrix for $A$ i.e., $A = L_A L_A^\top$     ▷ $O(n\tau^2)$, Lemma 2.9
13:       **while** $\|x_k - r\|_1 + \|y_k - c\|_1 \geq \epsilon_0$ **do**
14:          **if** $k \mod 2 = 0$ **then**
15:             $u_{k+1} \leftarrow u_k + \ln r - \ln(x_k)$
16:             $v_{k+1} \leftarrow v_k$
17:          **else**
18:             $v_{k+1} \leftarrow v_k + \ln c - \ln(y_k)$
19:             $u_{k+1} \leftarrow u_k$
20:          **end if**
21:          $x_k \leftarrow (\mathrm{diag}(e^{u_k})(L_A L_A^\top) \mathrm{diag}(e^{v_k}) + \mathrm{diag}(e^{u_k}) D \mathrm{diag}(e^{v_k})) \mathbf{1}_n$
22:          $y_k \leftarrow (\mathrm{diag}(e^{u_k})(L_A L_A^\top) \mathrm{diag}(e^{v_k}) + \mathrm{diag}(e^{u_k}) D \mathrm{diag}(e^{v_k}))^\top \mathbf{1}_n$
23:          $k \leftarrow k + 1$
24:       **end while**
25:       **return** $u_k, v_k, L_A, w$             ▷ We return $B(u_k, v_k)$ in a implicit way, i.e., $B(u_k, v_k) = \mathrm{diag}(e^{u_k})(L_A L_A^\top) \mathrm{diag}(e^{v_k}) + \mathrm{diag}(e^{u_k})(ww^\top) \mathrm{diag}(e^{v_k})$.
26: **end procedure**

---

**Theorem C.1** (Running time of Algorithm 4, Formal version of Theorem 4.2). *Given the cost matrix $C \in \mathbb{R}^{n \times n}$ with small treewidth $\tau$ and two simplex $r, c \in \mathbb{R}^n_+$, there is an algorithm (Algorithm 4) that takes $O(n\tau)$ for each iteration and $O(n\tau^2)$ for initialization to output*

- *a lower triangular matrix $L_A$*

- *vectors $u, v, w \in \mathbb{R}^n$*

*such that $B(u_k, v_k) \in \mathbb{R}^{n \times n}$ can be constructed (implicitly) by*

$$B(u_k, v_k) = \mathrm{diag}(e^{u_k})(L_A L_A^\top) \mathrm{diag}(e^{v_k})$$
$$+ \mathrm{diag}(e^{u_k})(ww^\top) \mathrm{diag}(e^{v_k}),$$

*satisfying*

$$\|B(u_k, v_k)\mathbf{1}_n - r\|_1 + \|B(u_k, v_k)^\top \mathbf{1}_n - c\|_1 \leq \epsilon_0.$$

*Proof.* The running time for each step is shown as follows:

- Writing down cost matrix $C \in \mathbb{R}^{n \times n}$ takes $O(n\tau)$ time as $\mathrm{nnz}(C) = n\tau$ by using Claim 2.11.

- Implicitly write down matrix $D \in \mathbb{R}^{n \times n}$, this takes $O(n)$ time since $D \in \mathbb{R}^{n \times n}$ is a rank-1 matrix.

- Initializing $x_0$ and $y_0$ takes $O(n\tau)$ as $\mathrm{nnz}(C) = n\tau$.

- Using Lemma 4.1, we know $L_A$ is $\tau$-sparse in column. Then, calculating the Cholesky decomposition for $A$ takes $O(n\tau^2)$ time using Lemma 2.9.

- Calculating $\mathrm{diag}(e^{u_k})(L_A L_A^\top)\mathrm{diag}(e^{v_k})$ takes $O(n\tau)$ time as $L_A$ is $\tau$-sparse in column.

- Calculating $\mathrm{diag}(e^{u_k})D\,\mathrm{diag}(e^{v_k})$ takes $O(n)$ time as matrix $D$ is a rank-1 matrix.

- Updating $u \in \mathbb{R}^n, v \in \mathbb{R}^n$ takes $O(n)$ time.

Hence, the initialization time for Algorithm 4 is $O(n\tau^2)$ and the per iteration running time is $O(n\tau)$. $\square$

## D  SYMMETRIC

---

**Algorithm 5** Sinkhorn's Algorithm for symmetric distribution with small treewidth

---

1: **procedure** SINKHORNALGORITHMSYM($r, \epsilon_0 \in (0,1)$)                    ▷ Theorem D.2
2:                                                                                 ▷ Accuracy $\epsilon_0$
3:     $k \leftarrow 0,$
4:     $u_0 \leftarrow 0$
5:     $v_0 \leftarrow 0$
6:     $w \leftarrow \mathbf{1}_n$
7:     $x_0 \leftarrow e^{-C_{i,j}/\gamma}\mathbf{1}_n$
8:     $y_0 \leftarrow \left(e^{-C_{i,j}/\gamma}\right)^\top \mathbf{1}_n$
9:     Implicitly form $D = ww^\top$
10:    Implicitly form $A \in \mathbb{R}^{n \times n}$, where $A_{i,j} = e^{-C_{i,j}/\gamma} - 1$
11:        ▷ Explicitly writing down $A$ requires $n^2$, however, we never need to explicitly write down $A$. Knowing the exact formulation of $A$ is enough to do the Cholesky decomposition
12:    $L \leftarrow$ Cholesky decomposition matrix for $A$ i.e., $A = L_A L_A^\top$        ▷ $O(n\tau^2)$, Lemma 2.9
13:    **while** $\|x_k - r\|_1 \geq \epsilon_0$ **do**
14:        $u_{k+1} \leftarrow u_k + \ln r - \ln(x_k)$
15:        $x_k \leftarrow (\mathrm{diag}(e^{u_k})(L_A L_A^\top)\mathrm{diag}(e^{u_k}) + \mathrm{diag}(e^{u_k})D\,\mathrm{diag}(e^{u_k}))\mathbf{1}_n$
16:        $k \leftarrow k+1$
17:    **end while**
18:    **return** $u_k, L_A, w$                          ▷ We return $B(u_k)$ in a implicit way, i.e.,
       $B(u_k) = \mathrm{diag}(e^{u_k})(L_A L_A^\top)\mathrm{diag}(e^{u_k}) + \mathrm{diag}(e^{u_k})(ww^\top)\mathrm{diag}(e^{u_k}).$
19: **end procedure**

---

In this section, we provide an algorithm (Algorithm 6) to solve the OT problem in $O(\epsilon^{-2}n\tau\|C\|_\infty^2 \ln n)$ time, given the two distribution are identical, i.e., $c = r$.

**Definition D.1.** *Given the symmetric OT problem* $\arg\min_{X \in \mathcal{U}_r}\langle X \rangle$*, we define*

$$\mathcal{U}_r = \{X \in \mathbb{R}_+^{n \times n} : X\mathbf{1}_n = r, X^\top \mathbf{1}_n = r\},$$

*where $\mathbf{1}_n$ is the all-ones vector in $\mathbb{R}^n$ , $C \in \mathbb{R}_+^{n \times n}$ is a given cost matrix, and $r \in \mathbb{R}^n$ are given vectors with positive entries that sum to one.*

We first provide the running time of the Sinkhorn's algorithm (Algorithm 5) for symmetric case.

**Theorem D.2** (Running time of Algorithm 5). *Given the cost matrix $C \in \mathbb{R}^{n \times n}$ with small treewidth $\tau$ and a simplex $r \in \mathbb{R}_+^n$, there is an algorithm (Algorithm 5) that takes $O(n\tau)$ for each iteration and $O(n\tau^2)$ for initialization to output*

- *a lower triangular matrix $L_A$*

- *vectors $u, w \in \mathbb{R}^n$*

*such that $B(u_k) \in \mathbb{R}^{n \times n}$ can be constructed (implicitly) by*

$$B(u_k) = \mathrm{diag}(e^{u_k})(L_A L_A^\top) \mathrm{diag}(e^{u_k}) + \mathrm{diag}(e^{u_k})(ww^\top) \mathrm{diag}(e^{u_k}),$$

*satisfying*

$$\|B(u_k)\mathbf{1}_n - r\|_1 \le \epsilon_0.$$

*Proof.* Similar to the proof of Theorem 4.2, here the two distribution are identical, i.e., $c = r$. □

---

**Algorithm 6** Approximate OT by Sinkhorn for symmetric distribution

1: **procedure** APPROXOTSYM($\epsilon$)                    ▷ Theorem D.4
2:                                                          ▷ Accuracy $\epsilon$
3:     $\gamma \leftarrow \frac{\epsilon}{4 \ln n}$
4:     $\epsilon_0 \leftarrow \frac{\epsilon}{8\|C\|_\infty}$
5:                        ▷ Find $\widetilde{r} \in \Delta^n$ s.t. $\|\widetilde{r} - r\|_1 \le \epsilon_0/4$ and $\min_{i \in [n]} \widetilde{r}_i \ge \epsilon_0/(8n)$.
6:     $\widetilde{r} \leftarrow (1 - \frac{\epsilon_0}{8})(r + \frac{\epsilon_0}{n(8 - \epsilon_0)}\mathbf{1}_n)$
7:     $(u, L, w) \leftarrow$ SINKHORNALGORITHM($\widetilde{r}, \epsilon_0/2$)    ▷ Algorithm 5
8:                        ▷ Note that $u, v, L, w$ is an implicit representation of $B$, i.e.,
   $\mathrm{diag}(e^u)(L_A L_A^\top)\mathrm{diag}(e^u) + \mathrm{diag}(e^u)(ww^\top)\mathrm{diag}(e^u)$
9:     $(p, X, Y, w, u) \leftarrow$ ROUND($u, L, w, r$)    ▷ Algorithm 7
10:    **return** $(p, X, Y, L, w, u)$  ▷ We return $\widehat{X}$ in an implicit way, i.e., $\widehat{X} := XBY + pp^\top/\|p\|_1$
11: **end procedure**

---

**Algorithm 7** Rounding of the projection of $B$ on $\mathcal{U}$ for symmetric distribution

1: **procedure** ROUNDSYM($u \in \mathbb{R}^n, L \in \mathbb{R}^{n \times n}, w \in \mathbb{R}^n, r \in \mathbb{R}^n$)    ▷ Lemma D.3
2:                        ▷ $L$ is a lower triangular matrix that only has $O(n\tau)$ nonzeros
3:                        ▷ We never explicit write $B$. $B$ can implicitly represented by
   $\mathrm{diag}(e^u)(L_A L_A^\top)\mathrm{diag}(e^u) + \mathrm{diag}(e^u)(ww^\top)\mathrm{diag}(e^u)$
4:     $X \leftarrow \mathrm{diag}(x)$ with $x_i = \min\{\frac{r_i}{(B\mathbf{1}_n)_i}, 1\}$
5:     $B_0 \leftarrow XB$                                 ▷ We only implicitly construct $B_0$
6:     $Y \leftarrow \mathrm{diag}(y)$ with $y_j = \min\{\frac{r_j}{(B_0^\top \mathbf{1}_n)_j}, 1\}$
7:     $B_1 \leftarrow B_0 Y$                              ▷ We only implicitly construct $B_1$
8:     $p \leftarrow r - B_1 \mathbf{1}_n$
9:     **return** $p, X, Y, w, u$          ▷ We return $G$ in an implicit way, i.e., $G := B_1 + pp^\top/\|p\|_1$
10: **end procedure**

---

Next, we show the running time of the rounding algorithm (Algorithm 7) for symmetric case.

**Lemma D.3** (An improved version of of Lemma 7 in (Altschuler et al., 2017))**.** *Given $r \in \triangle_n$, $B \in \mathbb{R}_+^{n \times n}$, $u \in \mathbb{R}^n$, there is an algorithm (Algorithm 7) that outputs*

- *a diagonal matrix $X \in \mathbb{R}^{n \times n}$*

- *a diagonal matrix $Y \in \mathbb{R}^{n \times n}$*

- *a lower triangular matrix $L_A$*

- *vectors $u, w, p \in \mathbb{R}^n$*

*such that $G \in \mathcal{U}_r$ can be constructed (implicitly) by*

$$\widehat{X} = X(\mathrm{diag}(e^u)L_A L_A^\top \mathrm{diag}(e^u) + \mathrm{diag}(e^u)(ww^\top)\mathrm{diag}(e^u))Y + pp^\top/\|p\|_1,$$

*satisfying*

$$\|G - B\|_1 \le 2(\|B\mathbf{1}_n - r\|_1),$$

*in $O(n\tau)$ time.*

*Proof.* Similar to the proof of Lemma A.5 and Lemma 4.4, here the two distribution are identical, i.e., $c = r$. □

Overall, we provide the running time of the algorithm (Algorithm 6) that approximate the OT for symmetric case.

**Theorem D.4.** *There is an algorithm (Algorithm 6) takes cost matrix $C = MM^\top = \in \mathbb{R}^{n \times n}$, an $n$-dimensional simplex $r$ as inputs and outputs*

- *a diagonal matrix $X \in \mathbb{R}^{n \times n}$*

- *a diagonal matrix $Y \in \mathbb{R}^{n \times n}$*

- *a lower triangular matrix $L_A$*

- *vectors $u, w, p \in \mathbb{R}^n$*

*such that $\widehat{X} \in \mathcal{U}(r)$ can be constructed (implicitly) by*

$$\widehat{X} = X(\mathrm{diag}(e^u)L_A L_A^\top \mathrm{diag}(e^u) + \mathrm{diag}(e^u)(ww^\top)\mathrm{diag}(e^u))Y + pp^\top/\|p\|_1,$$

*which satisfies Eq. (2) in*

$$O(n\tau^2 + \epsilon^{-2}n\tau\|C\|_\infty^2 \ln n)$$

*time.*

**Remark D.5.** *If we don't care about the output format to be lower-triangular matrix, then the additive term $n\tau^2$ can be removed.*

*Proof.* By using Theorem D.2, we have the running time of Line 7 is $O(n\tau \cdot T)$, where $T$ is the total number of Sinkhorn's algorithm iterations. By using Lemma D.3, the running time of Line 9 is $O(n\tau)$. The rest of the proof is similar to the proof of Similar to the proof Theorem B.3. □

## LLM USAGE DISCLOSURE

LLMs were used only to polish language, such as grammar and wording. These models did not contribute to idea creation or writing, and the authors take full responsibility for this paper's content.

