# OpenReview forum: "Faster Sinkhorn’s Algorithm with Small Treewidth"
_ICLR.cc/2026/Conference — ICLR 2026 Conference Withdrawn Submission_

### Official Review · Reviewer_dg48 · 2025-10-28

**Soundness:** 2
**Presentation:** 3
**Contribution:** 2
**Rating:** 2
**Confidence:** 3

**Summary:**

This paper proposes a faster algorithm for approximating **Optimal Transport (OT)** distances—specifically, entropy-regularized OT solved via **Sinkhorn’s algorithm**—by exploiting *small treewidth structure* in the cost matrix $\(C = AA^\top\)$.
While standard Sinkhorn variants run in $\(\tilde{O}(\varepsilon^{-2} n^2)\)$ time (Dvurechensky et al., 2018), this work shows that when $\(A\)$ has treewidth $\(\tau \ll n\)$, the runtime can be improved to $\(\tilde{O}(\varepsilon^{-2} n^\tau)\)$.

The key idea is to represent the exponential kernel $\(K = e^{-C/\gamma}\)$ implicitly through sparse Cholesky factors derived from $\(A\)$, avoiding explicit $\(n^2\)$-sized computations.
The paper reformulates Sinkhorn’s updates and the rounding phase to operate entirely in this implicit representation and proves convergence and runtime bounds.
This yields the first provable runtime improvement of Sinkhorn’s algorithm under bounded-treewidth structure.

**Strengths:**

- It has first proven runtime improvement of Sinkhorn’s method under small-treewidth structure.
- convergence and runtime proofs are complete and technically sound.
- implicit kernel representation via sparse Cholesky factors is clean and generalizable.
- applicable to structured domains (graphical models, power networks, structured vision).

**Weaknesses:**

- No experiments: the paper is purely theoretical; empirical evidence of runtime gain or scalability is missing.
- a large portion of the proof techniques are borrowed from previous works, and not very new

**Questions:**

.

---

> ### Author Response · Authors · 2025-11-22
> **Thank you for your review**
>
> The authors sincerely thank the reviewers for their time and thoughtful comments, and we appreciate the feedback provided.

---

### Official Review · Reviewer_v9Bx · 2025-10-29

**Soundness:** 1
**Presentation:** 1
**Contribution:** 2
**Rating:** 0
**Confidence:** 4

**Summary:**

This paper presents an algorithm for the Sinkhorn procedure that is claimed to run in linear time. The key idea is to exploit the fact that the cost matrix can be decomposed in low-rank form, $C = A A^\top$, where the matrix $A$ additionally has small tree width (i.e., sparse columns).

**Strengths:**

The topic of the paper is interesting, and the proof techniques appear to have some theoretical merit (the fact that $K$ has sparse Cholesky factor when $C = A A^\top$ with sparse $A$ is interesting, it show that it "inherits" from the structure of $C$, which could be leveraged).

**Weaknesses:**

Despite the potential relevance of the subject, the paper suffers from many major issues, which make it look more like a work-in-progress than a finished research article. For this reason, my review will be rather brief.

First and foremost, although the authors present an algorithm for solving entropic OT, there is absolutely no implementation and experimental section to support their claims. This point alone prevents me from recommending this paper for publication and strongly influences my overall score.

Second, there are several methodological issues. The paper completely overlooks a large and important body of related work. The use of low-rank structures to accelerate linear-time Sinkhorn algorithms is not new at all — see, for example, [1, 2]. These prior works are never discussed, put into context, or compared empirically.

Moreover,  the assumption that such a cost structure is available is also never discussed: in practice, it never holds, and it is not clear how one would obtain such a decomposition. Presumably, one would need a separate algorithm to approximate the cost matrix by one of this form, which would eliminate the claimed linear complexity once the full computational cost is considered.

Finally, the paper is poorly written. It contains numerous typos, reads more like a sequence of definitions and minor results than a coherent methodological paper, and fails to clearly highlight the main contributions.

For all these reasons, I believe the paper is not ready for publication in its current form.

[1] Massively Scalable Sinkhorn Distances via the Nyström Method, ltschuler, Jason and Niles-Weed, Jonathan and Rigollet, Philippe, NeurIPS, 2017.

[2] Linear Time Sinkhorn Divergences using Positive Features, Scetbon, Meyer and Cuturi, Marco, NeurIPS, 2020.

**Questions:**

See above

---

> ### Author Response · Authors · 2025-11-22
> **Thank you for your review**
>
> The authors sincerely thank the reviewers for their time and thoughtful comments, and we appreciate the feedback provided.

---

### Official Review · Reviewer_mbo5 · 2025-10-30

**Soundness:** 3
**Presentation:** 3
**Contribution:** 2
**Rating:** 2
**Confidence:** 4

**Summary:**

In this paper, the authors study the problem of approximating the optimal transport (OT) distance within an additive error of $\varepsilon$. They focus on accelerating the Sinkhorn algorithm, whose standard implementation requires $\tilde{O}(n^2/\varepsilon^2)$ time. The key assumption is that the cost matrix $C=AA^⊤$ has small treewidth $\tau$. Under this structural condition, $C$ admits a Cholesky factorization with only $O(n\tau)$ nonzeros. Although the corresponding kernel matrix $K=exp⁡(-C/\gamma)$ is dense, the authors show that all necessary computations can be carried out implicitly using the sparse factorization of $C$. This reduces the per-iteration cost of Sinkhorn from $O(n^2)$ to $O(n\tau)$, leading to a total runtime of $\tilde{O}(n\tau/\varepsilon^2)$ for obtaining an $\varepsilon$-approximation of the OT distance.

**Strengths:**

The paper establishes that, under a structural assumption of small treewidth in the cost matrix, the Sinkhorn algorithm for entropy-regularized optimal transport can be executed in sub-quadratic time.

**Weaknesses:**

The paper offers limited algorithmic novelty. It does not propose a fundamentally new algorithm or improved convergence analysis for Sinkhorn in the low-treewidth regime; instead, it simply observes that the standard Sinkhorn iterations can be implemented more efficiently when the cost matrix admits a sparse factorization. The authors also do not discuss concrete application domains where the cost matrix would have small treewidth, leaving the practical relevance of the assumption unclear. Moreover, for certain structured settings such as planar or grid graphs, existing separator-based or multiscale methods already yield faster or even exact optimal transport computations. Given that the main attraction of Sinkhorn is its practical value—through regularized solutions and GPU acceleration—such a paper should, in my opinion, include an empirical analysis demonstrating the benefit. Unfortunately, the authors do not do so.

**Questions:**

NA

---

> ### Author Response · Authors · 2025-11-22
> **Thank you for your review**
>
> The authors sincerely thank the reviewers for their time and thoughtful comments, and we appreciate the feedback provided.

---

### Official Review · Reviewer_UvhZ · 2025-11-02

**Soundness:** 2
**Presentation:** 1
**Contribution:** 2
**Rating:** 2
**Confidence:** 3

**Summary:**

The authors produce the first algorithm to compute entropy regularized transport plans on graphs with $n$ vertices and bounded treewidth $\tau$ in sub-quadratic time, improving upon existing $\varepsilon$-approximate algorithms running in $O(\varepsilon^{-2} n^2)$ time. To do this, they speed up the well-studied Sinkhorn scaling algorithm for entropy regularized optimal transport by implicitly maintaining the cost matrix using the Cholesky factorization. The Cholesky factorization has bounded complexity $O(n\tau)$ and requires $O(n\tau^2)$ time to compute. This, along with existing bounds on the convergence rate of the Sinkhorn algorithm, implies an algorithm computing entropy regularized OT on graphs with bounded treewidth in $O(\varepsilon^{-2}n\tau^2)$ time, significantly improving upon the prior quadratic complexity in $n$.

**Strengths:**

One of the main drawbacks of the Sinkhorn algorithm in practice is its quadratic complexity, and the authors successfully reduce the time complexity of running Sinkhorn in specific cases.

The paper is crisp and succinct, with no wasted space.

**Weaknesses:**

The fact that Sinkhorn's papers are not cited makes me question if the authors are aware of the necessary and sufficient conditions for convergence of Sinkhorn's algorithm proved in the 1960s. While everything in this work seems technically correct (mostly, with some small and easily fixed error discussed in questions), this omission raises the concern about its significance.

 The obvious limitation of this work is that it is only beneficial on graphs with bounded treewidth.


   Paper is not easy to read.  For instance, putting section 1.3 before section 2 makes all of the notation in section 1.3 confusing. The function $\phi$ isn't even defined until section 3, but is necessary to understand section 1.3. Furthermore, the statement of Theorem 1.1 uses the treewidth ``of a matrix'', which is not the standard use of treewidth and not properly defined until section 2.

**Questions:**

The  significance of the work is unclear. To my understanding, the claimed primary contribution of this paper is a proof that the Sinkhorn algorithm can be applied to graphs with bounded treewidth $\tau$ in $\widetilde{O}(n\tau)$ time.  I believe it is known that the Sinkhorn algorithm for matrix scaling converges if and only if the matrix has total support; see e.g. ``Concerning Nonnegative Matrices and Doubly Stochastic Matrices" by Sinkhorn and Knopp 1967. So the proof that Sinkhorn converges should be straightforward? If this is true, then the main contribution lies in the compressed representation of the transport plan which allows one to execute each iteration of Sinkhorn in $O(n\tau)$ time. But it is known by [BGHK95] that an implicit representation exists and can be computed quickly. Clarity on the authors' contributions would be much appreciated. It would be helpful if the authors clarify any other challenges  that they address in this work.
|
The authors claim on line 72 that ``in practical settings, treewidth tends to be small" and provide one example reference to power system analysis data. Do they have other examples to reference? When and why should one expect treewidth to be small in practice?  This is important because Sinkhorn often requires quadratic time ibecause t it is assumed one is given all $O(n^2)$ pairwise distances.
|
Can  the Cholesky factorization be computed efficiently in parallel. One of the main draws to Sinkhorn in practice is its simplicity and efficiency to run in parallel.

The cited reference for computing the Cholesky factorization takes $O(n\tau^2)$ time. Does this mean the whole algorithm actually requires $O(\varepsilon^{-2} n\tau^2)$ time instead of $O(\varepsilon^{-2} n\tau)$?

---

> ### Author Response · Authors · 2025-11-22
> **Thank you for your review**
>
> The authors sincerely thank the reviewers for their time and thoughtful comments, and we appreciate the feedback provided.

---

### Note · Authors · 2025-11-22

**Comment:**

The authors sincerely thank the reviewers for their time and thoughtful comments, and we appreciate the feedback provided.

**Withdrawal Confirmation:**

I have read and agree with the venue's withdrawal policy on behalf of myself and my co-authors.